# Estimates of Genetic Parameters for Direct and Maternal Effects on Pre-Weaning Growth Traits in Turkish Saanen Kids

**DOI:** 10.3390/ani13050940

**Published:** 2023-03-05

**Authors:** Funda Erdoğan Ataç, Çiğdem Takma, Yakut Gevrekci, Şeniz Öziş Altınçekiç, Tugay Ayaşan

**Affiliations:** 1Department of Animal Science, Faculty of Agriculture, Ege University, Izmir 35100, Turkey; 2Department of Animal Science, Faculty of Agriculture, Bursa Uludağ University, Bursa 16059, Turkey; 3Kadirli Faculty of Applied Sciences, Osmaniye Korkut Ata University, Osmaniye 80000, Turkey

**Keywords:** Turkish Saanen, growth traits, maternal genetic effect, heritability

## Abstract

**Simple Summary:**

It is economically significant to perform early selection of breeding goats. Given that the effect of genetic and environmental factors on growth can be estimated more accurately, estimation of adult body weight is viable, and rapid genetic progress can be achieved in the herd through the application of selection in this manner. Thus, profitability can be increased in goat breeding. Additionally, genetic and maternal influences on pre-weaning growth traits are important in identifying variation between individuals. In this study, herd, sex, age of dam, type and year of birth, and maternal genetic influence were considered according to their effects on pre-weaning growth characteristics in Turkish Saanen kids. As a result of the study, it has been determined that the economic significance of the maternal effect is high due to its effect on the growth characteristics of Turkish Saanen kids. In the selection program for offspring who grew up alongside their mothers from birth until weaning, it is recommended to account for the maternal effect as well as environmental factors.

**Abstract:**

Data on 4487 Turkish Saanen kids from 176 bucks and 1318 dam-goats, obtained from the Turkish Saanen goats in the Izmir region and collected between 2018 and 2019, were analyzed to examine the effect of genetic and non-genetic factors on growth traits. The average birth weight of the kids was determined as 3.33 ± 0.68 kg, the average W60 was 13.06 ± 2.94 kg, the average WW was 18.38 ± 4.14 kg, and the average PreWDG until weaning was 0.17 ± 0.04 g. Model 1, which does not account for the maternal effect, and Model 2, which includes the maternal effect, were used in the estimation of genetic parameters. The heritability estimates of BW, W60, WW, and PreWDG ranged from 0.05 to 0.59 in both models. It is recommended to consider the maternal effect as well as the environmental factors in the selection program for the best early breeder selection of kids growing alongside their mothers until the weaning period.

## 1. Introduction

The Turkish Saanen breed was developed by crossing the Saanen breed with native Turkish breeds (e.g., Saanen × Hair, Saanen × Kilis). Turkish Saanen goats are known for their high milk production and fertility efficiency and are commonly used in goat husbandry in Turkey, especially in Western Anatolia. The male kids are generally marketed for meat [1,2,3]. As a result, growth characteristics of goats are important biological factors that affect the sustainability and profitability of businesses, and these are economically significant for early breeding of breeding goats. In addition, improving growth performance is an important way of increasing meat output in a kid production system [4,5]. In particular, body weight and number of kids at birth are known to be associated with environmental factors as well as genetic traits [6,7]. The potential for genetic improvement is dependent largely on the heritability of the trait measured and its relationship with other traits of economic significance [8]. Birth weight is an important parameter of economic significance in animals due to its high and positive correlation with growth rate, weaning weight, and adult live weight, as well as its effect on the viability of newborn animals [9]. For this reason, it is necessary to examine the growth and morphometric characteristics of goats in order to evaluate their growth potential more accurately. Thus, profitability can be increased in goat breeding by incorporating these characteristics into breeding programs. In other words, to provide the development of an optimal breeding program, knowledge of parameters for growth traits is vital.

On the other hand, by revealing the effect of genetic and environmental factors on growth, a more accurate estimation of adult live weight is possible, and rapid genetic progress can be achieved in the herd with the application of selection in this direction [10]. Accurate performance evaluation and estimation of genetic parameters across various traits are prerequisites for creating successful selection strategies and breeding programs [11,12,13]. In addition, genetic and maternal effects on pre-weaning growth characteristics are important in defining variation among individuals [14,15]. Although maternal effects reflecting maternity ability are more important in the early stages of life, they have effects that can be transferred to later periods of life [16]. Especially in the period from birth until weaning, keeping the kids with their mothers makes the maternal effect even more impactful. Therefore, there are many studies that demonstrate that the maternal effect should be taken into account in order to reach more reliable genetic predictions in selection programs [13,17,18,19,20,21,22]. Although there are studies on the genetic parameters and factors affecting the growth characteristics before and after weaning in goats, there are not enough studies on the genetic parameters of growth characteristics in Turkish Saanen kids.

The goal of this study was to determine the most appropriate time, accounting for maternal effects, to determine early selection based on the growth characteristics of Turkish Saanen kids. Therefore, genetic parameters for pre-weaning growth traits in Turkish Saanen kids were estimated. Based on their effects on pre-weaning growth traits, herd, gender, dam age, type of birth and birth year, and maternal genetic effect were all considered.

## 2. Materials and Methods

### 2.1. Animals and Experimental Procedure

The dataset collected from the Turkish Saanen Goat subproject in the Izmir region as part of the National Sheep and Goat Improvement Project was examined in this study. A total of 4487 Turkish Saanen kids, including 176 bucks and 1318 dams, were studied. Growth trait records of Turkish Saanen kids were collected from 15 herds between the years 2018 and 2019. Birth weight (BW), 60th day live weight (W60), weaning weight (WW), and pre-weaning daily live weight gain (PreWDG) were employed as the growth traits in kids. The kids were fed alongside their mothers until weaning and were weaned on the 90th day.

### 2.2. Statistical Analyses

Herd, gender, type of birth, age of dam, and birth year effects on BW, W60, WW, and PreWDG of Turkish Saanen kids were analyzed. Analyses of variance components and heritability were carried out for BW, W60, WW, and PreWDG records. For this purpose, the model is structured with herd, gender, age of dam, type of birth, and birth year being the fixed effects; BW as a covariate effect for W60, WW, and PreWDG; and age of dam also as a covariate effect for BW. Animals’ additive genetic, maternal genetic, and residual effects were considered to be random effects. Descriptive statistics of the growth traits of Saanen kids are given in Table 1.

BW, W60, WW, and PreWDG were 3.33 ± 0.68 kg (CV = 20.51%), 13.06 ± 2.94 kg (CV = 22.51%), 18.38 ± 4.14 kg (CV = 22.51%), and 0.17 ± 0.04 g (CV = 26.89%), respectively (Table 1).

The following single trait and single record animal models were used:Y = Xb + Za + e(1)
Y = Xb + Za + Zm + e(2)
where, 

Y: a vector of BW, W60, WW and PreWDG; 

b: a vector of herd, gender, type of birth, age of dam, and birth year as fixed effects; BW as a covariate for W60, WW, and PreWDG; and age of dam also as a covariate effect for BW; 

a: direct additive genetic effects; 

m: maternal genetic effects; 

e: residual effect, X, and Z are incidence matrices relating observations to b, a, and m, respectively. 

The estimates of variance components for each trait were estimated using the Average Information-Restricted Maximum Likelihood (AI-REML) method by Meyer [23].

## 3. Results

For all growth traits, the effects of herd, gender, type of birth, and birth year were found to be significant (*p* < 0.01). On the other hand, the effects of birth type on PreWDG were not significant (*p* = 0.39). Therefore, all effects were included in the model to obtain estimates of variance components. When the flocks were examined, the average BW was 3.26 ± 0.68 kg, the highest BW was 3.63 kg in the 6th herd, and the lowest BW was in the 10th flock at 2.74 kg. It was observed that W60 averaged 13.06 ± 2.94 kg and ranged between 8.53–16.04 kg, and WW averaged 18.38 ± 4.14 kg and varied between 11.64–21.65 kg. The average daily live weight gain was 0.17 ± 0.05 kg. Male kids were found to be heavier than female kids in all growth stages (*p* < 0.01). It was observed that twin births were the most common in Turkish Saanen kids, followed by singleton and triplet births, and quadruplet births had the lowest rate of 3%. The weights of single-born kids were found to be higher than those of multiple-born kids in all growth periods. For BW, 5-year-old mothers were found to have heavier kids than mothers of other ages, and for W60 and WW, kids of 7-year-old mothers were found to be heavier. The effect of the year of kids on live weights was found to be higher and more significant in 2018 than in 2019 (Table 2).

Variance components of BW, W60, WW, and PreWDG traits that were affected by herd, gender, birth type, the age of dam, and birth years were estimated. The additive, phenotypic, residual variances and heritability estimations are given in Table 3. In Model 1, the estimates of additive genetic variance were the lowest for BW (0.02). The additive genetic variance estimates increased rapidly by age—they reached 2.77 for WW. Phenotypic variance estimates have also shown the same tendency as the additive genetic variances; while the variance was 0.43 in the BW, it was estimated to be 4.70 in the WW. In this research, major differences have been identified between the heritability estimations obtained from BW to WW for different ages. Estimates of heritability were found to be 0.05 ± 0.02, 0.57 ± 0.002, 0.59 ± 0.04, and 0.33 ± 0.002 for BW, W60, WW, and PreWDG, respectively, in Model 1.

In the present study, the maternal effect was not included in Model 1 of Turkish Saanen goats for all growth traits, and additive genetic variances ranged from 0.02 to 2.77. Additive genetic variances ranged from 0.02 to 0.43 in Model 2, in which the maternal genetic effect was also considered. Moreover, maternal genetic variances ranged from 0.001 to 0.22 in Model 2. Additive genetic variances and, therefore, direct heritability estimates caused overestimation—the heritability estimates obtained in the present study were 0.57 ± 0.002 and 0.59 ± 0.04 for W60 and WW in Model 1, respectively. The heritability estimate for PreWDG was lower with the maternal model than the direct animal model. PreWDG heritability estimates were lower in Model 2 (0.22 ± 0.001), where the maternal effect was better accounted for than in Model 1 (0.33 ± 0.002).

## 4. Discussion

Many studies have been conducted to determine the factors that influence live weight in kids at various stages of development. Thiruvenkadan et al. [24] stated that the effect of birth year, birth season, birth type, and gender are important. According to Mio et al. [25], mean daily body weight gain and WW weight are significantly higher (*p* < 0.01) in males than in females. Birth type was determined as effective on BW (*p* < 0.01); weaning age with WW as effective on daily body weight gain (*p* < 0.05), gender (excluding BW (*p* < 0.01)), WW (*p* < 0.05), and daily body weight gain (*p* < 0.01); and the season of birth as effective on BW, WA, and daily live weight gain (*p* < 0.01), excluding WW. Supakorn and Pralomkarn [19] found that the effect of birth year, birth type, and gender on WW was significant (*p* < 0.05). Atouı et al. [26] reported that birth year, birth type, gender, maternal age, and maternal weight had a significant (*p* < 0.05) effect on BW. Alade et al. [27] discovered the effects of genotype, birth type, and gender to be significant in all growth periods (*p* < 0.01). Cappai et al. [28] investigated the interpretation of the metabolic profile of the transition goat raised in an extensive farming system. They found that metabolic patterns related to pre-partum, post-partum, and single - twin gestation.

In this study, singletons and males had significantly higher body weights compared to multiples and females at all growth stages (*p* < 0.01). These results were found to be similar to the study by Tozlu and Olfaz [29], which reported that the effects of genotype, birth type, and gender on live weights at 30, 75, and 180 days were significant. On the other hand, it was stated that only genotype had a significant effect (*p* < 0.05) on the daily live weight gains of kids between birth and after 30 days, the viability of kids at the age of 30 days was significantly affected by gender, and the viability of males was higher than that of females (*p* < 0.01). Ghimir et al. [30] found the effect of genotype and gender on live weight to be significant (*p* < 0.05). They did not, however, find any significant effect of dam age or season on body weight or daily body weight gain at any period (*p* > 0.05). Similarly, they reported that WW and PreWDG were not affected by any factor. According to Alade et al. [27], Akdag et al. [31], and Tozlu and Olfaz [29], male kids were significantly heavier than females (*p* < 0.01) among singletons compared to those born with multiples. They found that kids born to mothers aged five years had the highest BW (*p* < 0.01). 

According to Atouı et al. [26], fixed effects, such as gender, species, and age of dam, which have significant effects on the body weight selection to be applied, should always be taken into account. The estimates of heritability for BW, W60, WW, and PreWDG in our current study ranged from 0.05 to 0.59. These results were higher than those reported by Rashidi et al. [11] for Markhoz goats (0.02), Kasap et al. [32] for Saanen goats (0.04), and Menezes et al. [33] for Boer goats (0.01). Moreover, our estimate of direct heritability for WW was higher than that estimated by Rashidi et al. [11] for Markhoz goats (0.03), Maghsoudi et al. [34] for Iranian Cashmere goats (0.07), and Mokhtari et al. [35] for Raeini Cashmere goats (0.15).

On the other hand, Otuma and Osakwe [36] found the estimated heritability values of BW and body weight at 90 and 360 days to be 0.41 ± 0.08, 0.45 ± 0.31, and 0.45 ± 0.28, respectively. They claimed that by selecting individuals with high body weights at a young age, they could achieve higher body weights at weaning age. According to Thiruvenkadan et al. [24], the heritability of live weight tends to increase from birth to age 160 days. Heritability estimates for pre- and post-weaning weight gain were found to be 0.29 ± 0.12 and 0.39 ± 0.17, respectively. Onder et al. [7] estimated the genetic parameters of sex and birth type, which are thought to be effective on live weights from birth to 180 days of age, and found the genetic variance to be 0.14 and heritability to be 0.27 for BW. It was stated that the selection to be made for any of these traits will also result in significant improvement for other traits.

Kuthu et al. [12] found heritability estimates for BW, W60, WW, and PreWDG of 0.28 ± 0.23, 0.26 ± 0.44, 0.23 ± 0.32, and 0.21 ± 0.32, respectively. They reported that the growth traits considered in selection should have low to moderate heritability and the growth traits should be based on estimated breeding values in selecting the animals that are future parents of the herd. Jawasreh et al. [37] determined their heritability estimates as 0.30 ± 0.04 for BW, 0.19 ± 0.04 for WW and PreWDG, and as 0.2 ± 0.04 for WW. Tesema et al. [13] reported their h2 estimates for BW, WW, and PreWDG as 0.38, 0.12, and 0.09, respectively. Koşum et al. [38] found heritability for BW and WW as 0.43 and 0.05, respectively. Gunia et al. [39] estimated heritability for live weight of Capricorn goats at 70 days and 11 months as 0.20 and 0.32, respectively. In their study, Rashidi et al. [11] estimated direct heritability to be at a moderate level of 0.21 for daily live weight gain, 0.22 for BW, and 0.16 for WW. Dashtizadeh et al. [40] estimated heritability as 0.55, 0.18, 0.47, and 0.43; Oseni and Ajayi [41] as 0.50–0.59, 0.14–0.4, 0.29, and 0.11, for BW, WW, and live weight at 90 and 180 days, respectively; and Mohammadi et al. [42] found values of 0.22, 0.25, and 0.29 for BW, WW, and W180. Bhattarai et al. [43] determined the heritability estimate for BW as 0.37 ± 0.12. 

In the present study, heritability estimates in growth traits were found to be higher in Model 1, in which maternal effect was not taken into consideration. On the other hand, in Model 2, where maternal effect was also included, estimates were observed to be lower. Therefore, model 2 resulted in lower, and possibly more realistic, variance components. Maternal effects play a significant role in the expression of pre-weaning traits. As a result of this, it is important to know the extent of both maternal genetic and environmental effects, in addition to the additive genetic effects. Hammoud and Salem [44] reported that maternal effects should be taken into consideration when carrying out genetic evaluations of pre-weaning growth traits. Similarly, Magotra et al. [45] highlighted the considerable role of maternal effects on early growth traits. Supakorn and Pralomkarn [17] found direct heritability estimates to be 0.44 and 0.51, and maternal heritability estimates as 0.15 and 0.16 for BW and WW, respectively, in kids of four different goat species. Supakorn and Pralomkarn [19] recorded direct heritability estimates as 0.26 and 0.38, and maternal heritability estimates as 0.09 and 0.12 for BW and WW, and highlighted that the most suitable model to achieve rapid progress in WW in the herd would be one that included maternal genetic effect and excluded the direct maternal genetic covariance. Bedhane et al. [46] found direct additive heritability, additive maternal heritability, and maternal environmental effects values of 0.04–0.39 and 0.02–0.08, 0.09–0.20 and 0.03–0.07, and 0.12–0.21 and 0.05–0.09, respectively. They reported additive genetic variance, maternal permanent environmental variance, and phenotypic variances for BW and WW as 0.02–0.05 and 0.09–0.22, 0.03–0.04 and 0.15–0.26, and 0.21–0.26 and 2.49–2.70, respectively. They stated that maternal environmental effects and maternal permanent environmental effects were critical variation sources for body weights of young kids. In addition, they suggested considering the non-genetic effects on growth traits in any breeding program. 

In the study in which they estimated the variance components and genetic parameters for birth and weaning weights, Gholizadeh et al. [18] found the maternal effect on the examined traits to be significant and suggested considering maternal effects in a future selection program. Barazandeh et al. [21] estimated genetic parameters of effective factors on BW, WW, and PreWDG, and reported that the model including additive direct genetic and permanent maternal environmental effects was the most suitable. Bangar et al. [15] and Bangar et al. [22] obtained low levels of heritability estimates for BW and WW. When compared with our study, these findings are similar in terms of BW, but higher than our findings for WW. Tesema et al. [13] recommended that both the direct additive genetic effect and maternal effects should be considered as environmental variation sources. Schoeman et al. [47] examined heritability estimates and the variance components of the factors that are thought to be effective on BW and WW and found that lower but more accurate variance components were obtained when maternal genetic effects were included in the individual model. Roy et al. [16] used models in which maternal genetic or maternal permanent environmental effects were excluded and included. According to the results of the study, heritability estimates were higher for all traits when maternal effects were ignored. They reported heritability estimates for BW and WW to be 0.12 and 0.18, respectively. Based on the heritability estimates obtained from the present study, it is possible to select kids prior to weaning for breeding.

## 5. Conclusions

In conclusion, the effects of environmental factors such as herd, birth year, gender, birth type, and age of dam were significant for all growth traits in Turkish Saanen kids. At the same time, the maternal effect is of great importance due to its effect on the growth characteristics of Turkish Saanen kids. As a result of comprehensive knowledge of these traits, it is possible to design the successful inheritance of economically significant traits in goat breeding. This information is considered an important step in the planning and implementation of any successful selection or breeding program aimed at improving the genetic gain of animals. While Turkish Saanen goats, of which there is a substantial population, are used for milk production, some of the offspring born are used for breeding and a significant amount are used for butchery. Therefore, it is recommended that early selection that considers environmental factors as well as maternal effects at the pre-weaning age may be delivered for genetic progress in Turkish Saanen kids growing with their mother from birth to weaning. This effect should also be considered in fertility programs. In addition to these, further studies examining the primary effect at the F2 level are required. 

## Figures and Tables

**Table 1 animals-13-00940-t001:** Descriptive statistics of the growth traits of Turkish Saanen kids.

Traits	Mean	Std. Deviation	Coefficient of Variation (%)
BW	3.33	0.68	20.51
W60	13.06	2.94	22.51
WW	18.38	4.14	22.51
PreWDG	0.17	0.04	26.89

BW: birth weight; W60: weight at 60th day; WW: weaning weight; PreWDG: pre-weaning daily gain.

**Table 2 animals-13-00940-t002:** Analysis of variance of non-genetic affecting growth traits in Turkish Saanen kids.

Factors	n	(BW) (kg)	W60 (kg)	WW (kg)	PreWDG (g/day)
		3.26 ± 0.68	13.06 ± 2.94	18.38 ± 4.14	0.17 ± 0.05
Herd					
1	588	3.42 ± 0.03 ^ab^	10.18 ± 0.10 ^f^	14.72 ± 0.14 ^h^	0.13 ± 0.002 ^g^
2	430	3.62 ± 0.03 ^a^	12.75 ± 0.11 ^cd^	16.60 ± 0.17 ^g^	0.14 ± 0.003 ^f^
3	44	2.99 ± 0.10 ^def^	8.53 ± 0.36 ^g^	11.64 ± 0.52 ^i^	0.10 ± 0.006 ^h^
4	23	3.55 ± 0.13 ^a^	11.21 ± 0.50 ^def^	17.24 ± 0.72 ^fg^	0.15 ± 0.008 ^cdefg^
5	141	3.34 ± 0.05 ^abc^	12.29 ± 0.20 ^de^	18.80 ± 0.29 ^bcde^	0.17 ± 0.003 ^bcde^
6	65	3.63 ± 0.08 ^a^	11.60 ± 0.30 ^e^	16.03 ± 0.43 ^g^	0.14 ± 0.005 ^fg^
7	245	3.08 ± 0.04 ^cde^	13.28 ± 0.15 ^bc^	18.46 ± 0.22 ^def^	0.17 ± 0.003 ^bcde^
8	44	3.21 ± 0.10 ^bcd^	12.96 ± 0.36 ^bcde^	16.82 ± 0.52 ^g^	0.15 ± 0.006 ^ef^
9	115	3.43 ± 0.06 ^ab^	9.79 ± 0.22 ^fg^	14.74 ± 0.32 ^h^	0.13 ± 0.004 ^g^
10	134	2.74 ± 0.06 ^g^	12.29 ± 0.21 ^de^	19.02 ± 0.30 ^cd^	0.18 ± 0.004 ^bc^
11	225	3.40 ± 0.04 ^ab^	13.29 ± 0.16 ^bc^	21.65 ± 0.23 ^a^	0.20 ± 0.003 ^a^
12	56	2.89 ± 0.09 ^a^	16.04 ± 0.32 ^a^	19.87 ± 0.46 ^c^	0.19 ± 0.002 ^ab^
13	1051	3.53 ± 0.02 ^bcde^	15.23 ± 0.07 ^a^	21.23 ± 0.11 ^b^	0.19 ± 0.002 ^a^
14	654	3.16 ± 0.03 ^cde^	13.81 ± 0.09 ^b^	18.68 ± 0.14 ^d^	0.17 ± 0.002 ^bcd^
15	668	3.09 ± 0.02	12.61 ± 0.09 ^de^	18.20 ± 0.13 ^ef^	0.17 ± 0.02 ^de^
*p*-value		0.01	0.01	0.01	0.01
Gender					
Male	2046	3.39 ± 0.04 ^a^	13.08 ± 0.14 ^a^	18.71 ± 0.21 ^a^	0.17 ± 0.002 ^a^
Female	2441	3.12 ± 0.04 ^b^	12.47 ± 0.14 ^b^	17.73 ± 0.21 ^b^	0.16 ± 0.002 ^b^
*p*-value		0.01	0.01	0.01	0.01
Type of birth					
Single	1099	3.61 ± 0.04 ^a^	13.32 ± 0.14 ^a^	18.80 ± 0.21 ^a^	0.17 ± 0.002
Twin	2482	3.28 ± 0.04 ^b^	12.99 ± 0.13 ^ab^	18.53 ± 0.20 ^ab^	0.17 ± 0.002
Triplet	765	3.05 ± 0.04 ^c^	12.49 ± 0.15 ^ab^	17.99 ± 0.22 ^b^	0.16 ± 0.002
Quadruplet	141	3.08 ± 0.06 ^c^	12.36 ± 0.22 ^b^	17.57 ± 0.34 ^b^	0.16 ± 0.004
*p*-value		0.01	0.01	0.01	0.39
Age of dam					
1	639	3.19 ± 0.04 ^a^	12.81 ± 0.13 ^a^	18.11 ± 0.19 ^ab^	0.16 ± 0.002 ^cd^
2	1315	3.22 ± 0.03 ^bc^	12.55 ± 0.11 ^c^	18.03 ± 0.17 ^c^	0.16 ± 0.002 ^de^
3	977	3.28 ± 0.03 ^b^	12.31 ± 0.11 ^bc^	17.56 ± 0.17 ^c^	0.16 ± 0.002 ^e^
4	835	3.24 ± 0.03 ^bc^	12.46 ± 0.12 ^bc^	17.59 ± 0.18 ^bc^	0.15 ± 0.002 ^de^
5	364	3.30 ± 0.04 ^bc^	12.70 ± 0.15 ^bc^	18.51 ± 0.22 ^a^	0.17 ± 0.002 ^abc^
6	173	3.19 ± 0.05 ^cd^	12.49 ± 0.19 ^ab^	17.95 ± 0.28 ^a^	0.16 ± 0.003 ^ab^
7	104	3.27 ± 0.07 ^bcd^	13.14 ± 0.23 ^abc^	19.20 ± 0.35 ^a^	0.18 ± 0.004 ^a^
8	80	3.25 ± 0.08 ^d^	12.78 ± 0.28 ^abc^	17.71 ± 0.42 ^abc^	0.16 ± 0.005 ^bcde^
*p*-value		0.01	0.01	0.01	0.01
Birth year					
2018	1987	3.37 ± 0.04 ^a^	13.99 ± 0.15 ^a^	19.62 ± 0.22 ^a^	0.18 ± 0.002 ^a^
2019	2500	3.14 ± 0.04 ^b^	11.60 ± 0.14 ^b^	16.83 ± 0.21 ^b^	0.15 ± 0.002 ^b^
*p*-value		0.01	0.01	0.01	0.01

Values with different letters in the same row are different (*p* < 0.01).

**Table 3 animals-13-00940-t003:** Variance components and heritability values of growth traits.

	**Model 1**					
Traits	$\sigma_{a}^{2}$		$\sigma_{p}^{2}$		$\sigma_{e}^{2}$	$h_{a}^{2}$ ± S.E.
BW	0.02		0.43		0.41	0.05 ± 0.02
W60	0.42		0.73		0.31	0.57 ± 0.002
WW	2.77		4.70		1.92	0.59 ± 0.04
PreWDG	0.31		0.94		0.62	0.33 ± 0.002
	**Model 2**					
Traits	$\sigma_{a}^{2}$	$\sigma_{p}^{2}$	$\sigma_{e}^{2}$	$\sigma_{m}^{2}$	$h_{a}^{2}$ ± S.E.	$h_{m}^{2}\;$± S.E.
BW	0.02	0.43	0.41	0.001	0.05 ± 0.02	0.002 ± 0.018
W60	0.43	0.86	0.32	0.13	0.50 ± 0.005	0.13 ± 0.002
WW	0.11	1.05	0.70	0.22	0.11 ± 0.003	0.22 ± 0.002
PreWDG	0.10	0.94	0.62	0.22	0.11 ± 0.001	0.22 ± 0.001

BW: birth weight; W60: weight at 60th day; WW: weaning weight; PreWDG: pre-weaning daily gain; $\sigma_{p}^{2}:$ phenotypic, $\sigma_{a}^{2}$: additive genetic, $\sigma_{m}^{2}:\mathrm{maternal}\;\mathrm{genetic},$ and $\;\sigma_{e}^{2}:$ residual variances, $h_{a}^{2}$: heritability of direct additive; $h_{m}^{2}$: heritability of maternal genetic effects; and S.E.: standard errors.

## Data Availability

All datasets collected and analyzed during the current study are available from the corresponding author on fair request.

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
