# Peer review of "Estimates of Genetic Parameters for Direct and Maternal Effects on Pre-Weaning Growth Traits in Turkish Saanen Kids"

_animals, 2023, doi:10.3390/ani13050940_

Round 1
Reviewer 1 Report
The manuscript aimed to determine the most appropriate time, considering maternal effects, to determine the early selection in terms of growth characteristics 70 of Turkish Saanen kids.
The authors studied several parameters for pre-weaning growth traits (the herd, gender, damage, type of birth and birth year, and maternal genetic effect ) in 71 Turkish Saanen kids.
The word Genotype has been mentioned several times in the Discussion, but it's unclear what a breed or specifically selected breeding line is.
Please, use "breed" instead of "race".
Too many references from non-indexed publications were used in the Discussion. Please, modify your work in the publication list.
Since some pre-weaning growth traits show a statistically significant influence of the maternal effect on them, why is this factor still not introduced into breeding schemes? The authors have not mentioned the maternal effect on F2 studied elsewhere. Have such studies been conducted so far?
I recommend a significant revision of the manuscript.
Author Response
First of all, thank you for your review.
- Since the word "genotype" was used by the authors in the reviewed literature, it was used in the same way in the discussion.
- In many sources, it is seen that the goats created as a result of crossbreeding, as in Saanen goats, are named after that country, and when it has a widespread effect, it has become one of the genetic resources of that country over time, without a determination such as blood level. In this context, 3830 studies can be found by searching "Turkish Saanen Goats" in google scholar. And it was observed that most of these studies were published in regularly published peer-reviewed journals with high impact factor.
- An explanation about the Turkish Saanen breed was added to the introduction.
- Current literatures were added.
- It was emphasized in the study that maternal effect should be included in breeding programs. In this study, some reproductive parameters such as birth type were also taken into account indirectly. The importance of including maternal effect in breeding programs was added to the conclusion section. However, no study was found that specifically examined the effect of maternal effect on F2.
- This manuscript was edited by editing services to the standards of professional.

Reviewer 2 Report
Comments from Reviewer
Line 45-46
and these are economically important for early breeding of breeding goats[sounds like a repetition-consider revising]
Line 80-81
Growth trait records were composed of 4487 Turkish Saanen kids that came from 176 bucks, 1318 dams, and a total 1318 from 15 herds between 2018 and 2019[revise sentence to allow for ease of comprehension]
Line 94
Put table on a separate page
Line 132
Table 2: Compress in one page
Line 161
Many studies are conducted to determine the factors influencing live weight in kids at various stages of development[language structure for discussion-consider revising]
Line 162 -168
What is the relationship of these studies with the present study in Turkey Saanen goats? Discuss the relationships
Line 225-226
Did the model used have an influence on the values??
General Comments
The text is generally not a smooth read. The style of the discussion is not a smooth read, could the author revise the writing style and also refer to the general flow in the whole manuscript.
Suggestion: Review
[Ma1]Sounds like a repetition, consider revising the sentence
[Ma2]Revise sentence, the meaning seems lost
[Ma3]Consider revising the language use in discussions….
Author Response
First of all, thank you for your review.
- Line 45-46: The sentence “the use of the proper selection program in Turkish Saanen goats is critical to the Turkish economy” removed from the manuscript.
- Line 80-81: The sentence was corrected.
- Line 94: It will be done through the editor at the stage of publication.
- Line 132: It will be done through the editor at the stage of publication.
- Line 161: The sentence was corrected.
- Line 162-168: Current literatures were added.
- Line 225-226: Yes. The models used in the manustript led to different variance components and heritability estimates for the growth traits examined. And these results was presented and discussed in Table 3.
- This manuscript was edited by editing services to the standards of professional.

Reviewer 3 Report
Dear authors,
thank you for this piece of work Which I Found meritorious of being considered to be processed for publication I found interest in it and I think that your paper deserves attention by the readership.
It is opinion of this reviewer that the English form should be proofread extensively for it is difficult to read.
Overall one major flaw consists in the writing style.
Additionally this reviewer believes that the identification of factors suffers from the description of the animal side effect which in turn means that additional information should be provided as to the physiologic parameters leading to the metabolic background determining the differences pounded out through your models.
For instance management and pregnancy conditions had to be described in detail. As a matter of fact single vs twin pregnancies play a role on birth weight of kids. I believe that the reference to the paper published by Cappai et al., 2019 research in veterinary science, 123: 84-90, doi:10.1016/j.rvsc.2018.12.016 is worth of notes in this context as a reference to your data.
As a consequence, discussion and conclusion may benefit of the comparative approach.
Author Response
- First of all, thank you for your review. Considering the suggestions, the manuscript was edited by editing services to the standards of professional.
- Metabolic parameters were not used in this study. However, as seen in the literatures used in this study, heritability estimates were obtained without using metabolic data. In our future research, it is planned to use measurements including metabolic data.
- Suggested literature was added to the discussion.

Round 2
Reviewer 1 Report
The revision is fairly well-made and the comments were addressed adequately. The manuscript has been improved.
I recommend the manuscript for publication.
Author Response
All recommendations were done.
Our article has been edited by a translation company.

Reviewer 3 Report
Dear Authors,
Thank you for this revised version of the manuscript and for welcoming all the points I raised and change the text accordingly.
I have just few minor comments. The simple summary still continues to be too long.
Additionally, most of the amended tracts in the text need to be proofread for english style.
The paper can be then accepted for publication in my opinion.
Author Response

(The authors gave the same response as above.)
